# Conflicting HIV/AIDS Sex Education Policies and Mixed Messaging among Educators and Students in the Lower Manya Krobo Municipality, Ghana

**DOI:** 10.3390/ijerph192315487

**Published:** 2022-11-22

**Authors:** Benedict Ocran, Sharon Talboys, Kimberley Shoaf

**Affiliations:** 1Department of Social Work, Care and Community, School of Social Sciences, Nottingham Trent University, 50 Shakespeare St., Nottingham NG1 4FQ, UK; 2Division of Public Health, School of Medicine, University of Utah, 50 North Medical Drive, Salt Lake City, UT 84108, USA

**Keywords:** HIV/AIDS education, Sub-Saharan Africa, comprehensive sex education, abstinence education, young people, Ghana

## Abstract

While school-based comprehensive sex education (CSE) is effective in HIV prevention among young people ages 10–24 years, Ghana’s national sexual and reproductive health education policy promotes abstinence. Meanwhile, the Ministry of Health’s HIV prevention programs provide more comprehensive school-based education. This qualitative study evaluated the HIV/AIDS education program in the Lower Manya Krobo Municipality to assess the perspectives of students and educators in 10 schools on school-based sexual and reproductive health programs, including HIV/AIDS education and conflicting HIV/AIDS sex education policies. HIV prevalence in the Lower Manya Krobo Municipality of Ghana was more than twice the national average at 5.64% in 2018, and prevalence among youth in the municipality aged 15–24 was the highest in the nation at 0.8%. Educators have mixed feelings regarding abstinence-based and CSE approaches. However, students generally endorse abstinence and describe the limitations of condom use. Ambiguity in overarching policies is identified as a factor that could influence the orientation of school-based health educators, create disharmony in sex education interventions, introduce confusing sex education messages to young people, and create a potentially narrow curriculum that limits the gamut of HIV/AIDS sex education to exclude young people’s risky sexual behaviours and diverse teaching and implementation strategies. Policies and the scope of sex education should be realigned to ensure the transparent implementation of HIV/AIDS sex education programs in Ghana.

## 1. Introduction

Globally, school-based HIV/AIDS education programs have significantly prevented HIV infection among young people aged 10–24 years through behaviour change communication methods [1,2,3]. The Global Campaign for Education in 2004 estimated that expanding Universal Primary Education (UPE), including providing behaviour change communication, could prevent 700,000 new HIV infections annually [4]. This is especially important in Sub-Saharan Africa (SSA), where the region accounts for 82% (1.5 million) of the 1.6 million young people worldwide affected by HIV/AIDS [5].

Research has shown that school-based HIV/AIDS programs that focus on comprehensive sex education (CSE) are generally effective compared to those that focus on abstinence-only education. Comprehensive sex education (CSE) is an age-appropriate, culturally relevant approach to teaching sexuality and relationships by providing scientifically accurate, realistic, non-judgmental information [6]. CSE has also effectively increased voluntary HIV testing and reduced early and unintended pregnancies in Africa [7]. Kirby et al. [8] also argue that school-based CSE programs effectively reduce teenage pregnancies and STDs through increased knowledge of prevention strategies. A review of school-based sexuality education programs suggests that interventions that include content about the role of gender and power dynamics in sexual relationships are more efficacious in HIV education [9]. When presented without judgement and including social aspects of power and negotiation, CSE empowers young people to make better sexual and reproductive health choices and negotiate aspects of intimate relationships [6,9]. Indeed, CSE is at the heart and centre of HIV/AIDS education in the attempt to locally implement international development goals on the sexual and reproductive health needs of young people and HIV/AIDS [7].

Abstinence-based modes of sex education are defined as programs that exclusively promote abstinence (from premarital sex) and do not provide information regarding condoms and contraception [10]. In contrast to comprehensive forms of sex education, there is evidence to support that abstinence-based programs have little positive influence on sexual and reproductive health outcomes, including HIV/AIDS. Kirby [9] argues from a review of 56 abstinence and comprehensive sex education programs that abstinence-based education neither delays sex nor has any positive effects on sexual behaviour. Similarly, a systematic review of abstinence-based programs and HIV/AIDS has shown little evidence of abstinence-based education preventing HIV/AIDS infection [11].

Despite the substantive evidence of the benefits of CSE in HIV/AIDS education, the processes associated with the design and implementation of CSE curricula in many contexts, including SSA, is culturally challenging and politically conflicting. Yankah and Aggleton, in their deliberation on the process that went into developing the CSE technical guidelines under the UN, noted that CSE curriculum design is a political as well as a technical process [12]. This political process is charged with local religious and cultural opposition to the CSE agenda [12], which makes its implementation in many countries daunting.

Interestingly, Ghana hosts a basic school system where abstinence-only and CSE interventions are implemented concurrently [13,14,15,16,17,18,19]. School-based sex education programs are detailed in overarching policy documents such as the *School Health Education Program* of the Ghana Education Service [20], the *National HIV/AIDS and STI policy* under the Ghana AIDS Commission [21], and the *Adolescent Reproductive Health Policy for young people in Ghana* [22]. Under the guidance of these documents, the Ministry of Health provides technical support in implementing sexual and reproductive health education programs through school-based interventions such as the *Enhanced School Health Education program* and “*HIV Alert*”, an HIV/AIDS program [14,23] that draws on the Abstinence, Be Faithful and Condom Use (ABC) approach, which is, in fact, a USAID foreign policy [24,25]. The ABC approach was developed by the (US) President’s Emergency Plan for AIDS as a midway between abstinence-based education and CSE [24]. However, in Ghana, the ABC is supported under the Government’s program on abstinence [26,27]. In effect, the Ghana Education Service, under its School Health Education Policy, stipulates that sex education interventions are to be aimed at the “*prevention of premarital sex*” [20], formally limiting all discussions on sexuality and relationships to the confines of abstinence.

At the same time, the Ministry of Health supports and implements CSE programs in schools. Under this arrangement, officials from the Ministry of Health, such as nurses, are allowed into schools to implement CSE programs that aim to: “*Enhance and promote reproductive health; Increase contraceptive prevalence through the promotion of, access to and quality of family planning services; Develop and implement cross-cutting measures to ensure access and quality of reproductive health services; and Enhance and promote community and family activities, practices and values that improve reproductive health*” [28]. This arrangement is in place, despite the Ministry of Education’s stance on abstinence [20,29,30].

Religious and cultural norms largely drove a recent rejection of a CSE curriculum in 2019 in Ghana among traditional and indigenous groups who prefer abstinence-based education [14]. Furthermore, while supportive of the abstinence agenda, foreign-funded policies such as the ABC approach may be described as ambiguous in their implementation in Ghana. ABC also thrives on condom use, which is a CSE approach; therefore, its alignment with the abstinence agenda of the Ghanaian Government gives credence to the argument we put forward regarding the ambiguous nature of overarching sex education policies. This political climate presents a negative potential to introduce conflicting messages that could confuse young people’s sexual and reproductive health worldview concerning HIV/AIDS and could contribute to the rising infection rates in young people.

In addition to the challenges posed by dual modes of sex education, a review of current policies suggests a limited implementation of the comprehensive sex education content in school-based programs because it is not a standalone subject. A report by Alan Guttmacher on CSE programs in Ghanaian basic schools shows that CSE excludes critical topics such as negotiation skills, assessing and using contraceptives, and gender-based violence and human rights [31]. This limitation results from the inclusion of such vital topics only in electives at the senior high school level.

From the above, it can be noted that dual modes of HIV Education, which are competing, limit the gamut of HIV/AIDS education to address cultural norms, which are important considerations in HIV/AIDS education design and implementation. We also argue that competing curricula limit the ability of the Ghanaian government and its various ministries to have essential, and possibly life-saving, discussions at the highest levels, which would ensure consistent messaging regarding HIV prevention throughout schools in the country. Some research has highlighted the presence of dual modes of sex education curricula in the Ghanaian Basic school system [14,15,27,32]. However, few have attempted to identify the dual modes as *competing and problematic*. Further, less have focused on the consequences competing curricula pose to HIV/AIDS education program efficacy, particularly in the context of HIV/AIDS prevention among young people.

### Study Purpose

This study was undertaken to describe school-based HIV/AIDS education in junior high schools in the Lower Manya Krobo Municipality (LMKM) from the perspectives of students and municipal and school-based health education coordinators. Understanding the views of young people is an essential step in designing sex education curricula and HIV/AIDS education in Ghana [33,34]. This paper was motivated by the first author’s research elsewhere in Ghana, which shows the influence of competing sex education curricula on the ability of teachers to deliver sex education [14,27].

Specific areas of inquiry aimed to explain:The approach and content of HIV/AIDS sex education.Perceptions of the effectiveness of abstinence-only and comprehensive sex education.Teaching/learning preferences and recommendations for improvements among students and health educators.

In 2018, HIV prevalence in the general population of LMKM was 5.56%, more than twice the national average of 1.69%. In 2018, the prevalence in young people aged 15–24 was 0.8%, the highest in the nation [21]. Incidence in young people increased from 5.30% in 2013 to 5.56% in 2016 [1]. According to the Ghana AIDS Commission, the high prevalence of HIV among young people is due in part to decreased HIV campaigns in recent years [21].

Considering the high prevalence of HIV infections among young people in the district, of which a majority, approximately 54%, are enrolled in junior high schools [35], this study joins in the call for accelerated efforts to conduct school-based HIV/AIDS education programs and research [2,3,36,37,38]. Investigating student and teacher perspectives towards HIV/AIDS education in a sentinel of high HIV prevalence is essential to understanding how factors such as the local Krobo culture of the LMKM underpin perceptions towards dual modes of sex education.

## 2. Materials and Methods

Qualitative research methods included face-to-face key informant interviews with students and school-based health educators using a semi-structured interview guide. Ten of thirty-two public junior high schools in the LMKM of the Eastern Region of Ghana were selected with assistance from the Municipal Health Education Coordinator (MSHEC). The selection of schools was limited to public junior high schools because the sex education program under evaluation is only implemented in public schools. The LMKM was divided into sub-regions based on already existing subdivisions of the education district called circuits. This ensured geographic representation; schools were randomly selected within each sub-region.

Participants (key informants) were identified in each school, including school-based health educators (SBC) and students. School-based health coordinators were selected due to their direct involvement as facilitators of HIV/AIDS programs in the schools. A municipal-level health educator was also selected. Students trained as peer educators in the HIV alert program were considered for inclusion. Each school had several student peer educators, from which one male and one female were randomly selected as student participants for in-depth interviews.

BEO, a sexual health researcher, an indigene from the municipality and then a research assistant at Ensign College of Global Heath (now Ensign Global College), conducted data collection. Drawing on BEO’s status as an indigene, participants were offered the choice to conduct interviews in either the Krobo local language or English. All participants, including students, opted for English, but the Krobo language was used intermittently to facilitate an easier understanding of certain words or terms. These include ‘sex education’, ‘sexuality’, ‘gender’, and ‘cultural norms’. BEO also used his familiarity with the Krobo local language and translation of complex terms to interpret some of the participants’ perspectives in fulfilment of our argument to describe and explain the participants’ views in fulfilment of the study’s objectives.

Informed consent was obtained from the municipal and school-based educators before scheduling their interviews. Parental permission was sought through a letter to parents or caregivers, informing parents of the purpose of the study and the protection of student interests. Some teachers also served as guardians for students who lived away from parents and relatives in rented accommodations. Thus, such teachers could approve of student interests without parents. In this case, teachers’ approvals for the inclusion of student participants (and protection of student interests) were issued in the presence of students. In addition, assent for participation in interviews was also sought from students. Adult participants (municipal and school health education coordinator and teachers), parents, and guardians (teachers) of students were provided copies of signed consent forms. In contrast, students were provided copies of signed assent forms.

KS, a Professor in the Division of Public Health of the University of Utah, the affiliate institution of Ensign Global College in Ghana, was placed in charge of the ethical approval process and attended online sessions organised by the Internal Review Board of the University of Utah as part of the Ethics application process. Ethical approval was granted by the Institutional Review Board of the University of Utah, with approval number IRB 00097500, on 25 May 2017. All data were stored on a password-protected computer, and the researchers stored hard-copy document forms securely.

Data from fieldwork were transcribed, coded, and analysed by BEO to arrive at themes related to the structure and content of HIV/AIDS education in junior high schools and the perspectives of students and teachers on the effectiveness of HIV/AIDS programs. Transcribed and analysed data were not returned to participants to validate the findings due to the combined use of Krobo and English Language where necessary to facilitate data collection. Still, both second authors cross-checked the quotes of participants and the final themes to ensure that quotes supported the trend of discussions. Responses were analysed and guided by recurrent themes related to the study’s objectives. Themes regarding HIV/AIDS education content were coded per interview questions on the HIV/AIDS curriculum (recurrent themes related to the content of HIV/AIDS education). Perceptions of young people on the strengths and weaknesses of HIV/AIDS sex education activities approach to HIV/AIDS education (e.g., abstinence vs. CSE) and suggestions for improvement were elicited from all participants. Template analysis [39] was employed to correlate and triangulate common themes across the responses of educators and students. Themes from teachers and students were aggregated and summarised.

### Study Site

The LMKM, which hosts the Agormanya sentinel site, is one of 26 administrative districts in the Eastern Region of Ghana. It covers a total land mass of 1476 km. The main occupation of the local people is fishing and farming [40]. Agormanya is reported to have recorded the first HIV infection in the country [41]. In addition, the site has mostly recorded the highest HIV infection prevalence since countrywide monitoring of HIV infections began across all sentinel sites in 1992 [41,42], with the only exceptions being 2011 and 2014 [41]. Several factors are argued to account for the country’s consistently high rates of infections. These include frequent sex trade, irregular use of condoms, multiple sex partners, and early sexual debut among young people [41]. The socio-cultural and economic history of the LMKM provides a better understanding of the trends in HIV infections presented above. Arable lands were lost due to the British colonial government’s investment in palm oil exports in the 19th Century. In addition, post-independence in 1957, there were poor resettlement plans for indigenes after claiming local lands to construct the Akosombo Dam [41]. Similarly, local folk’s fishing areas were destroyed due to dam construction. Adopting local lands for national purposes economically threatened the people’s fishing and farming activities, pushing indigenes to migrate to other African countries, such as Nigeria and Cote D’Ivoire, for greener pastures and engagement in the commercial sex trade [41]. Indeed, the prevalence of the pandemic in the municipality is reported to have started with the return of indigenes who had engaged in commercial sex activities elsewhere and perpetuated the same activities within the municipality [41].

Poverty means different things for people living with HIV/AIDS (PLWHAs) in the municipality. Owusu [43] reports in a study into the poverty livelihoods of PLWHAs in the Municipality indicates that the PLWHAs have a high dependency burden of child dependents, including an increased number of orphans from HIV/AIDS and children from single parents who are also considered orphans. Lund and Agyei-Mensah [42], in a study into the role of Queen Mothers (women traditional leaders in the traditional leadership system in Ghanaian culture) and the traditional society in providing care for HIV/AIDS orphans and vulnerable children in the municipality, indicate that the extended family system, which could provide primary care such as food, shelter, and education for orphans from HIV/AIDS and single parents, is threatened. These findings by Owusu and Lund and Agyei-Mensah suggest parenting trends and a family support system which make young people vulnerable to HIV.

In the economic context provided above, evidence suggests that local cultural norms place young people vulnerable to HIV. Indigenes are particularly patrilineal, meaning that men have a higher socio-cultural status than women [44]. This implies that young women who are reported to have multiple sex partners may enter into consensual relationships with older men or men of the same age to satisfy valued needs and could be disadvantaged in terms of power relations, leading to reported irregular use of condoms [43]. Consensual activities also occur between older females and younger men in the municipality. For instance, a mixed-methods study into how power dynamics shape the sexual initiation of 215 male youth found that more than half experienced sexual debut before age 16. Of this number, more than 42% consisted of consensual sex activities between older women and younger men, with older women using verbal encouragement, gifts, and alcohol to achieve coercive sexual consent [45]. Next to Kumasi, Agormanya has the second-highest level of consensual activities between young men who have sex with men (MSM) and older men [46]. Over here, the sexual health of young men is threatened for the same reasons as young women who engage in consensual activities, which is to satisfy valued needs. Indeed, the high rate of sexual activities between young and older MSM in LMKM reflects consensual activities between younger and older MSM at the national level [45,46].

Finally, recent statistics show that 92% of indigenes are Christian (Catholic, Anglican, Lutheran, Pentecostal, and Charismatic) [35]. In addition to the contextual issues enumerated above, the abstinence plan underpinning Christian values, which disagrees with HIV prevention approaches such as condom education, suggests the need to interrogate educators’ and young people’s perspectives on school-based HIV prevention strategies in the municipality. This is critically important in light of seminal research which suggests that teachers’ personal values and beliefs vary the extent to which they teach sexuality education which is currently implemented in government/public schools under subjects such as religious and moral education, social studies, and science [14,27,31]. (Abstinence sex education is incorporated in subjects such as religious and moral education in state-funded schools. Please see refs. [14,31] for a more detailed description of the mode of implementing sex education curricula in Ghanaian State funded basic schools. Further details on the structure of HIV/AIDS (sex) education are also highlighted in Table 2 of this study).

## 3. Results

A total of 31 interviews were conducted with 20 students, 10 school-based health coordinators, and 1 MSHEC (as shown in Table 1). As indicated in Table 1, student ages ranged from 11–19, and most (65%) were in the JHS 3 grade level. Only 25% lived with their parents. This indicates how families from more remote areas send their children to live with other family members or board in rented accommodations to access junior high education. While the 75% of students living on their own cover both boys and girls, teachers were especially concerned for girls: ‘*Most of the students are sent to live on their own in rented apartments in compound houses. Some also live with their grandparents or family members. I find this quite worrying for the [small] girls since they lack parental control to guide their way of life and even provide properly for them*’ (SBC I).

Most SBCs/MSHEC were between the ages of 25 and 40, and most had earned a bachelor’s degree.

### 3.1. Approach and Content of HIV/AIDS Education

The Municipal School Health Education Coordinator (MSHEC) reported that teachers are expected to teach abstinence according to the Ghana Education Service Policy and curriculum guidelines [20]. Responses from school-based coordinators (SBCs) suggest that teachers and nurses who implement HIV/AIDS education programs primarily teach abstinence, with an occasional addition to the discussion of condom use. SBC J stated, ‘*we (teachers) have been using the ABC … we stress the A (Abstinence) and B (Being faithful) … and do away with C (Condom usage)*’. SBCs described that other stakeholders are also involved in sex education in schools, particularly nurses, who mention condom use as an option due to their perception of high rates of sexual activity among adolescents in the district. Only one teacher made explicit reference to *comprehensive sex education (CSE)* by stating, ‘*We promote the abstinence, but those from health teach the comprehensive … the usage of condoms*’ (SBC G). Educators often described the use of condoms as ‘comprehensive’ but did not offer evidence of other CSE components.

Despite the policies focusing on abstinence-based sex education, some teachers use their discretion to teach condom use to prevent HIV infection based on the recognition and perception of sexual activity among adolescents in the district. Succinctly put by SBC A, ‘*It is in only a few circumstances that we mention the use of condoms to prevent HIV because we realise the pupils here are sexually active*. SBC H affirms this by noting that even though they teach abstinence, the health workers ‘*… insist on condom usage due to the sexual activity of the young people so that they take precaution*’.

Students corroborated SBCs by reporting that educational content focuses primarily on abstinence as the best way to prevent HIV but sometimes covers condom usage. A female student stressed that HIV/AIDS education in school ‘… *is to alert students to abstain from sex and reduce the number of people who have AIDS*’ (female student, School H). This is echoed by a male student in School G who stated that HIV/AIDS education ‘… *helps we the children not to participate in sexual immorality* (to) *… prevent us from acquiring STDs’*. A male student from school A stated that *‘normally school tells us to abstain …* (however) … *in our clubs, they say that if you can’t abstain use condoms*’. A female student from the same school (A) said that ‘*when (outside) people come into the school, they talk about the protective measures’*. This suggests that teachers teach abstinence-based content while more comprehensive topics, such as condom use, are discussed in clubs and presented by guest presenters.

Specific HIV content described by participants included modes of transmission, means of prevention, the ABC approach, common misconceptions, and stigmatisation. A strong emphasis seems to be placed on chastity and moral behaviour and is sometimes covered in religious and moral education. According to the MSHEC, the HIV/AIDS content of the school health curriculum includes the causes of HIV, mode of transmission, and means of prevention. Some respondents also mentioned content focused on abstinence and chastity. The MSHEC reported that they usually dwell on the ABC approach, which emphasises abstinence and being faithful, before moving to C, which is condom usage. An SBC for School E stated that ‘… *some subjects (Religious and Moral Education) touch on chastity and abstinence and STIs*’ (SBC E).

In addition, some SBCs stated that some topics in the HIV curriculum, particularly outside the classroom, were misconceptions about the disease and stigmatisation. SBC E explained that transmission and stigmatisation are discussed by saying ‘… *how they can acquire it, what it is about, then stigmatisation’*. Further, the SBCs reported that the topics of abstinence and chastity also formed a significant part of the curriculum.

From the student perspective, the modes of acquisition and prevention of the disease constituted a significant part of the HIV curriculum. A female student from School B elaborated further by opining that the HIV/AIDS program ‘… *teaches about the disease and what you can do to acquire and prevent it’*. The female student for school D summarised the prevention strategy of HIV education by noting that,’ … *they teach us HIV/AIDS in schools so we will abstain from sex and other sicknesses and remain chaste until we marry*’. Chastity, as noted here, was further explained by another female student to include relationships. To elaborate, she stated, ‘*The HIV/AIDS education program teaches us how to protect ourselves and not to be doing boyfriend and girlfriend*’. Furthermore, a student illustrated the content of the HIV curriculum to include ‘*stopping sex and what to do to behave (well) in society*’.

### 3.2. Modes of Delivery and Implementation

In line with the HIV and AIDS policy of the Ministry of Education to promote comprehensive education at all levels [47], responses from the MSHEC and teachers showed that HIV/AIDS activities are integrated by various ways in school activities and programs. HIV/AIDS is primarily integrated into some subjects in the regular curriculum, such as religious and moral education, physical education, social studies, and science. In addition, there is a component of HIV/AIDS education in extra-curricular activities. These include worship, where poems on HIV/AIDS are recited, peer education and club meetings, where plays and drama on HIV/AIDS are enacted, and talks provided by nurses during school hours.

Students also described several ways through which HIV/AIDS programs were implemented. Almost all students referred to HIV/AIDS content in subjects such as physical education, religious and moral education, science, and social studies. Other respondents mentioned school assemblies, individual teacher gatherings or meetings with students, club meetings, posters in schools, health talks, worship (religious activity held on certain days of the week before resumption of classroom teaching and learning activities), and outreach programs by students as implementation strategies employed by the schools.

Outside organisations, such as NGOs, supplement school-based education. However, a health educator stated that ‘*there are currently no organisations involved in school-based HIV education activities due to lack of funding … as compared to previous years where international organisations such as USAID, Plan Ghana, UNICEF and other local NGOs such as the Planned Parenthood Association of Ghana (PPAG) were involved in SE activities in the Junior High Schools*’. On the other hand, SBC responses showed that health workers, such as nurses, were involved in sex education. As put by the SBC for School E, ‘*Apart from the Education Office, the one I know is Child Protection, but they don’t do sex education and let’s say the hospitals, specifically the nurses*’.

Further student analysis also shows that nurses from the Municipal Hospital occasionally visited for HIV/AIDS and sex education talks with the school community. However, female students in school A confirmed the involvement of a local-based NGO called the ‘No Yawa’ (‘No Yawa’—A local jargon literally translated ‘No Problem’) Club. She explained, ‘*The nurses, the community health nurses come around … One time, the No Yawa Club came around*’.

### 3.3. Perspectives on Abstinence and Comprehensive HIV/AIDS Education

The MSHEC and SBC interviewees showed a high preference for abstinence and tolerance for ‘ABC’ education but no evidence of awareness of or support of full CSE. Despite educators’ awareness that students become pregnant and are at risk, they were generally reluctant to recommend more comprehensive education. Some comments suggest that more comprehensive education corrupts students or promotes risky behaviour. For example, one SBC reported that abstinence is the most effective approach because comprehensive education is like ‘… *telling them they are free to go in for sex’* (SBC H). Notwithstanding, a female SBC for School G recommended that ‘*comprehensive* (sex education) *is most effective because as much as you teach abstinence, they still indulge*’. Similarly, a female teacher for School A recommended condom usage because, in her view, ‘*most students are sexually active with little or no parental control*’.

Most SBCs acknowledged that students are highly sexually active and that abstinence sex education was not effective enough in addressing the sexual health needs of young people.

While SBCs widely adopt abstinence-based education, most perceive it to be ineffective, especially considering the prevalence of teenage pregnancies. SBCs described three main reasons as the reason for the failure of abstinence-only education: (1) the migration of students to more urban areas for better study opportunities, (2) a lack of parental oversight alongside irresponsible parenting trends, and (3) consensual and non-consensual sexual activity among students with older men.

For example, SBC B said, ‘… because of the way the community is most of them (students) are living alone … they rent rooms for them in the towns, and the parents live in the villages’. Subsequently, SBC B continued to explain that ‘…. (she) *would have gone in for abstinence* (but is) *challenged by some students who live on their own … and they might go in for sex*.’ She explained, ‘… *there are instances where the girls may not go in for sex but may be forced by a man who agrees to satisfy their needs*’.

Students expressed support for abstinence as an approach to HIV/AIDS sex education, with a few suggestions for condom usage for those who cannot abstain. A male student for School A expressed support for abstinence simply because ‘*when you abstain, you won’t get the virus*’. Another student emphasised abstinence as a preventive approach over sharp objects as a mode of contracting the virus. So, for some students, including Female students, in School A, abstinence is the best. She, however, interjects that ‘… *some students can’t abstain … So for those who can’t abstain, they should teach them how to use condoms’*.

Other students showed interest in chastity, mainly because of the instructive influence it has on student choices. For the male student of School C, ‘*Chastity explains more about that (HIV/AIDS) … it says that we should not practice sex before marriage and one can be safe*’. Male student for school E also agreed by noting that ‘*Chastity educates us more about HIV and how to avoid the disease*’.

On the topic of condoms, students seemed to perceive that condoms are ineffective due to their potential to fail. The following statements highlight this point:

‘*If they abstain, they cannot get HIV … but for condoms, sometimes it can burst’ (female student, School A) ….‘Condom is artificial, so it can easily tear, or something can happen, and you can easily get HIV’ (male Student, School A) …. ‘During sexual intercourse, because a condom is a rubber, it can get torn’ (male student, School B) …. ‘The other method I don’t like is the condom usage because sometimes if you don’t know and you wear it, and you have sex, the condom can burst, and you can have it*’.(male Student, School C)

Still, on condom usage, other students explained that protection is inadequate because they do not know how to use them properly, ‘*because we those in the school are not supposed to use condoms*’. Simply put, ‘*we are not qualified to do that’* (male Student, School C).

### 3.4. Teaching Modalities and Learning Preferences

Some educators reported that HIV education could be best imparted by combining two or more teaching methods within and outside the classroom. A male teacher illustrated this point by providing examples such as charts, fliers, role play, and audio-visual. His reason for the support of two or more methods of teaching is because ‘… *the kids like watching movies so makes it more concrete and sticks*’. The male coordinator for School J also said in a few words that ‘*practical and demonstration is the best means’.* These suggestions highlight that a combination of methods imparted more concretely on student interests in HIV/AIDS education.

Some teachers also supported classroom-based HIV activities. The coordinator for School D explained that ‘*instructional hours’* is the best because ‘*students take it more seriously’*, meaning that using formal classroom time rather than other methods is more effective. SBCs described a preference for programs which allowed for greater student participation. The MSHEC provided examples of peer education sessions ‘… *Because they enjoy doing it and while doing it they learn’*. A female coordinator for School C also agreed that HIV activities ‘… *are most effective when they (children) are involved*’.

Others suggested that small group student discussions were more effective than large gatherings. A female coordinator (School F) explained her support of small groups by stating that ‘… *because of the large numbers, they are reluctant to talk’*. She provided an example: ‘*In the HIV Alert group, they were only 20 and very interested in the program*’.

Several students also referred to the teaching and learning of HIV/AIDS content as embedded in some subjects in the classroom as a practical approach to HIV/AIDS sex education. A reason provided by a female student of school H is that, ‘*In the classroom, we are many, so everybody brings his (or her) idea*’. In support, a female student for School D also explained, ‘*During class, all the teachers know, so it is interesting’.*

In addition, while programs rolled out in class appealed to students, others preferred nurses because, as stated by a female student from School I, ‘… *they explained it more than the teachers*’. Another female student explained that group gatherings initiated by teachers with girls or boys only keened her interest in HIV activities. ‘*For example’*, she further explained, ‘*On Fridays*, *they gather all girls in one class and educate them*’. Finally, clubs were favoured by some students because ‘*A lot is learnt ‘during such meetings*’ (male student, School D).

‘Talks’ or didactic lectures were described as the least effective method because they are not very engaging. The MSHEC explained this by saying, ‘*they normally give them talks day in and day out … so peer education sessions with practical initiatives make them like the talks better*’. SBC C echoed this assertion by emphasising that ‘… *at their level when to see it (content being imparted) it seems concrete for them*’. SBC D again stated that discussion is less effective since ‘*children are interested much in what they see’.*

Several students also expressed disinterest in HIV/AIDS education activities during worship and large gatherings. Reasons included the non-comprehension of HIV/AIDS-related poems recited during prayer. For a student, poem recitals during worship are not appropriate ‘… *because some of them you don’t understand*’. Again, the seeming lack of time for in-depth discussion into HIV content came up. A female student illustrated this point by noting that during worship, ‘*we don’t have enough time to discuss … you can’t say enough during worship*’. Shyness was also described as why students do not participate in discussions during worship. A male student (School J) explained that he could not contribute during worship ‘*because sometimes I feel shy to talk*’. Finally, health talks were not interesting for a male student (School J) because ‘… *they don’t go into details about SE’*.

### 3.5. Recommended Improvements by Educators and Preferences of Students

A summary of the perceptions of students and educators is provided in Table 2. In general, students and teachers offered similar responses regarding the approach and content of the course. Students preferred small group discussions and hands-on demonstrations. They enjoyed co-curricular activities, while educators would like to see a dedicated curricular system on the topic and more parental involvement. Students provided several recommendations.

A majority of SBCs and the MSHEC stressed the need for more attention to be given to HIV education and that it be more formally incorporated into the curriculum. To that extent, some teachers suggested that HIV/AIDS be handled as a subject so that ‘… *you don’t talk about it only when it appears in the subject*’ (SBC C). In that same discussion, another coordinator suggested that at least two periods a week for the teaching of HIV/AIDS be allocated. In this case, HIV/AIDS will receive the required attention ‘… *because if you have it on the timetable, come what may, you will remember to teach it*’ (SBC D).

Furthermore, there were suggestions for more advocacy with HIV programmes and increased stakeholder involvement in design and implementation. The MSHEC emphasised the need to retrain and monitor peer educators and SBCs, with a further suggestion to include girls’ education coordinators due to their relevance in the subject matter. SBC J shared a similar opinion by calling for increased stakeholder involvement because when ‘… *the children see new faces, they are pleased*’. Several calls also came to the participation of the media, the church, and parents.

SBC A also shared her view on the need to involve the constituency of parents with children living on their own in urban areas in HIV/AIDS school-based intervention design and implementation. She further suggested the need ‘… *to talk to parents to at least get a family member in whose care they can be left*’. She explained the need ‘… *to encourage parents to address the needs of students, especially girls*’ (SBC B).

Some students shared the same views with SBCs and the MSHEC on increasing HIV/AIDS education activities. The female and male students for schools C and D said that programs should be organised frequently, and teachers must continuously teach HIV/AIDS content. Correspondingly, there were some suggestions for workshops for teachers so that they could explain HIV/AIDS-related matters more.

Some students also believed that clubs and peer education could be encouraged so that ‘*we can educate our peers who can go to the house to educate about HIV*’ (male student, School H).

Regarding HIV content in classroom teaching and learning, a male student from School H suggested that teachers should ‘*explain it very well’*. To enhance the teaching of HIV activities during school worship and make it more interesting, a female student from School I referred to the old nature of teaching materials and suggested that more manuals and handouts on HIV should be provided to make it more interesting.

Finally, a female student suggested that teachers call students individually (rather than publicly) to advise them on identified issues concerning their sex and relationships. She reported, ‘*Sometimes female teachers call individuals and confront before the whole school … but they should contact the student individually and talk to them*’.

Some students also provided varying opinions on the issue of condom usage. A male student from School A proposed that ‘*by demonstrating how to use the condoms’,* student perception regarding the failure of condoms may be countered. A female student from the same school agreed by stressing that ‘*they have to teach them the process of wearing it properly*’.

The students also shared similar interests as SBCs in clubs and having community partners’ involved in HIV/AIDS activities in the school. A female student from School C stated, ‘*We have been hearing our teachers every day, so if someone comes from outside to talk about it, the students will be thrilled*’.

## 4. Discussion

It is clear that school-based educators generally adhere to the abstinence-based sex education policies prescribed by the Ministry of Education but do sometimes address condom use. However, they do not seem to venture into other CSE topics such as power dynamics, negotiation in relationships, gender differences, and stigma, nor do they employ a non-judgemental approach due to the strong emphasis on purity and morality. However, components of CSE are provided through other means, such as talks and counselling by visiting nurses, discussions in peer education groups, and occasionally talks during non-curricular worship and guests from NGOs. While sexual education may expand beyond abstinence education in classes such as science, social studies, and religious and moral education may occur, details are sparse.

Four major themes were identified in this inquiry: (1) a disjointed approach to policy implementation for educators hurts teacher orientation, and (2) students are receiving incomplete information and mixed messages. This appears to lead to beliefs by the students that could lead them to take unnecessary risks, such as not using condoms because they “aren’t qualified” to do so. (3) Community-school participation needs to address cultural factors related to adolescents’ sexual health. (4) Varied perspectives, modes, and teaching methods must be addressed in the school’s social environment.

### 4.1. Disjointed Policy Implementation and Teacher Orientation

Health-promoting policy derives from the overarching national policy on school health. As a reflection of misaligned national policies allowing for CSE and abstinence, school-based coordinators in the participating schools implement HIV/AIDS sex education programs based on abstinence as prescribed by the Ghana Education Service [20]. At the same time, other stakeholders, such as nurses, introduce more comprehensive forms of sex education, such as condom usage [28]. Furthermore, while some teachers focus on abstinence, others acknowledge that they believe students are engaging in risky sexual behaviours and agree that condom usage should be taught. According to UNESCO [6], school-based coordinators play a significant role in HIV/AIDS and sex education. The variation in opinions by educators, along with abstinence and CSE, is likely influenced by misaligned Ministry of Health and Ghana Education Service policies and is probably affected by individual preferences. In light of HIV/AIDS education and sex education efforts, variation in preferences mirrors disunited efforts in implementing sex education curricula and the propensity to limit the impact of HIV/AIDS sex education initiatives in the schools. In other cross-sectional research on CSE curriculum implementation in Ghana, Kenya, Peru and Guatemala, such preferences have prevented teachers from discussing CSE topics identified to be controversial [17].

Next, we also identified participants’ overwhelming subscription to the ABC approach as evidence of the ambiguous nature of sex education policies. This could be because many educators (and students) exhibited shifting positions by agreeing with the abstinence and faithful approach but also aligning with condom use.

The study, therefore, identified the ambiguity in overarching policies stipulating abstinence-only and comprehensive forms of sex education as the revolving factor for the different preferences of teachers, which threatens to affect their orientation and create disharmony in sex education interventions in schools. This observation is consistent with findings from other studies in Ghana that external factors such as unclear policy stipulations and cultural influences affect teacher orientations in facilitating CSE programs [14,27]. The underlying factor revealed in these studies is that abstinence remains the official approach to sex education in schools, thereby limiting the possibility or ability and willingness of facilitators such as school-based coordinators to implement CSE programs in the schools adequately.

### 4.2. Individual Skills and Incorrect Sex Education Messages

The Comprehensive National Strategic HIV and AIDS Strategic Plan for 2016–2020 presented by the Ghana Aids Commission put forward comprehensive knowledge on condoms to combat the disease among young people [48]. Further, the strategy highlights the importance of condom use as dual protection against HIV/AIDS and early and teenage pregnancy [48]. In this regard, young people’s views show a preference for condom use, while at the same time, proffering concerns associated with condom failure challenges the aim to increase condom use among young people by 90% among females and 95% among males [21]. Cultural and religious factors may account for students’ mixed preferences for and against condom use. To explain, the official GES curriculum teaches abstinence. In support, all respondents (100%), as a reflection of the religious stratification of the Municipality, are Christian [40,41], and Christian values support the abstinence agenda of GES. However, condom education, which runs parallel with the abstinence agenda, is taught in schools by nurses. It seems, therefore, that in the context of ambiguous sex education policy, the cultural–religious background of the students, which runs parallel to condom education, may be a factor accounting for student reservations towards condom use. The literature is replete with studies in Ghana [17,49,50] and other parts of SSA [51,52,53] on cultural and religious barriers to sex education.

The mixed responses of students towards condom use are problematic, especially given the literature which suggests that sexual intercourse is the highest mode of transmission of HIV in Ghana [54], and condom use is a very effective means of combating the virus [2,3]. Going by the same argument in the previous section, this study identifies young people’s mixed messages on condom use as a manifestation of limitations associated with ambiguous overarching policies stipulating abstinence-only programs. Clarity is therefore required in overarching policies to ensure that confusing messages do not influence the life skills of young people to be gained from HIV/AIDS interventions.

### 4.3. School-Community Links and Young People’s Risky Sexual Behaviour

The introduction of incorrect sex education messages, sparse information on gender power and sexuality, the perceived lack of parental oversight in the affairs of young people, the perception of consensual sexual activities between young girls and older men, and existing evidence of consensual activities between young boys and more aged MSM or older females, place young people in participating schools at a high risk of contracting HIV.

Self-reports from young people show that most live independently away from their parents for education. In addition, teachers also reported that some young girls are engaged in consensual sexual relationships with older men, thus agreeing with other studies on the high-risk sexual behaviours of young girls at the junior high school level in Ghana [15,18,55]. However, the reference of teachers only to the risky sexual behaviour of young girls introduces a gendered expectation of sexual behaviour, where abstinence is expected of girls but recommended for boys. This gendered notion seems to reflect the highly patrilineal thinking of Krobo culture, which informs how women are viewed in matters of sex and relationships, and to a large extent, dictates how women should behave sexually [43]. This finding by Owusu [43] is consistent with research on the intersectionality of cultural norms and expected risky sexual behaviours of young people in Ghana [49,56] and South Africa [53]. Meanwhile, there is glaring evidence of high-risk sexual behaviour between young boys, older men, and older females in the municipality [45,46]. Thus, the gendered focus of teachers on girls’ risky behaviour may be limiting in any attempt to comprehensively address the dangerous behaviour of both boys and girls who live on their own and are potentially vulnerable to HIV.

Further, the extended family system in the Krobo culture, which could have provided support for students staying with family members, is weakened [43,44]. This suggests a parenting trend which puts young people living on their own at risk of the pandemic. Meanwhile, the HIV/AIDS sex education curriculum in the schools does not provide adequate information on aspects of CSE such as contraception (condoms) and negotiation skills in power and gender and power under sexual relationships, which will help young people to negotiate risky sexual relationships [7]. While this calls for an enhanced CSE curriculum, there is a need to strengthen school and community links [57] with parents to identify ways to protect the sexual health of young people who find themselves away from home.

### 4.4. Diverse Preferences and Schools’ Social Environment

Under the school’s social environment, the World Health Organization stresses quality relationships between students and staff, adequate role modelling provided by health personnel who visit the school, and an inclusive environment in which all students are valued and preferences respected [57,58]. Multiple actors, including facilitators (educators, nurses, and NGO representatives) and beneficiaries (students), are involved in the implementation in the participating schools. In the context of CSE and abstinence, it is therefore expected, as seen in the findings, that the myriad of factors associated with CSE and abstinence approaches to sex education will vary preferences for the best practices for teaching within and outside the classroom. The multiple factors also result in different perspectives on how sex education activities should be incorporated into the curriculum and implemented. This also informs the preferences of students on facilitators of sex education programs.

To adequately address all these preferences and inclusiveness in the provision of sex education programs, it is essential to identify all these factors under the school’s social environment in designing HIV programs. This way, HIV/AIDS education in the schools will harness local resources in the school’s social environment [58], tailoring interventions to the myriad of preferences under CSE and abstinence.

## 5. Conclusions

This paper has examined educators’ and young people’s views on multiple HIV/AIDS sex education approaches. Findings show that most young people live independently in rented accommodations or with guardians or family members. Further, teachers also seem to pay particular attention to the risky sexual behaviours of young girls compared to young boys. Findings also demonstrate that the HIV/AIDS curriculum is implemented with abstinence-only and more comprehensive approaches incorporating condom usage. In correlation with overarching policies, the curriculum in the schools is based on topics such as abstinence, chastity and the avoidance of relationships and condom usage. Educators often described the use of condoms as “comprehensive” but did not offer evidence of other CSE components, which suggests they perceive that the ABC approach is comprehensive. Findings also show that although HIV/AIDS is not a subject, it is incorporated into the curriculum and co-curricular school activities by teachers and nurses from outside the school through different teaching and learning methods.

Students and teachers reported a preference for HIV/AIDS education focusing on abstinence, such as chastity. Still, they were open to condom usage, acknowledging that some students are sexually active and, therefore, at risk. Students also bought abstinence as the preferred approach and suggested condom usage, but at the same time, proffered many limitations associated with the failures of condom use. Educators and students also reported a variety of preferences for teaching and curriculum implementation.

Four issues were identified to influence educators’ and students’ engagements with the HIV/AIDS curriculum under abstinence and CSE approaches: disjointed sex education policy and teacher orientations to sex education delivery; incorrect sex education messages as a result of a misalignment between GES policies and Christian religious values supporting abstinence on one hand and condom education on the other; community-school links and a culturally influenced, gendered expectation of young people’s risky sexual behaviours where a slight emphasis is placed on young girls’ risky sexual behaviour, and which prescribes abstinence of girls but recommends for boys; the schools’ diverse social environment.

The study identified ambiguity in sex education policies as a possible threat to the orientation of school-based coordinators as facilitators in sex education programs, a cause of disharmony in the implementation of sex education programs, and the introduction of confusing sex education messages for young people.

To ensure transparent implementation of HIV/AIDS sex education programs, policies and scope on abstinence-only and comprehensive forms of sex education should be realigned to enhance the orientation of school-based coordinators, incorporate cultural and religious norms which prevent harmony of sex education messages for young people, incorporate risky sexual behaviours of young girls and boys, and address the variety of preferences for the curriculum and the way it is to be implemented. Future research should evaluate the involvement of parents in HIV/AIDS sex education programs in the municipality to ensure a higher level of stakeholder participation in the design and implementation of sex education interventions.

### Limitations

The evidence provided in the literature on the risky sexual behaviours of young people (young boys’ sexual activities with men and young girls’ consensual activities with older men) was not self-reported by young people but by teachers (reports of sexual activities between young girls and older men). This suggests a certain degree of social desirability bias from the self-reports provided by students, probably due to the cultural tendency of students to present themselves as sexually inactive. However, we are confident that, to a large extent, teacher perspectives on student (sexual) behaviour lessen the social desirability bias that could have challenged the validity of student data elicited and usually challenges conclusions derived from program evaluations [59].

## Figures and Tables

**Table 1 ijerph-19-15487-t001:** Socio-demographic characteristics of interview respondents in Lower Manya Krobo Municipality, Ghana (N = 31).

Students (N = 20)		School-Based and Municipal Health Education Coordinators (N = 11)	
	n (%)		n (%)
**Age**		**Age**	
11–14	9 (45)	25–30	4 (36)
15–19	11 (55)	31–35	4 (36)
		36–40	2 (18)
		41–45	0 (0)
		46–50	1 (9)
**Sex**		**Sex**	
Female	11 (55)	Female	4 (36)
Male	9 (45)	Male	7 (64)
**Religion**		**Religion**	
Christianity	20 (100)	Christian	11 (100)
Others	0 (0)	Other	0 (0)
**Current Grade Level**		**Education Level**	
JHS 1	3 (15)	Diploma	2 (18)
JHS 2	4 (20)	Bachelor	9 (82)
JHS 3	13 (65)	Masters	0 (0)
**Living With**			
Parents	5 (25)		
Relatives	4 (20)		
Others	11 (55)		

JHS (Junior High School).

**Table 2 ijerph-19-15487-t002:** Summary of student and health educator perspectives on sex and HIV education in junior high schools, Lower Manya Krobo Municipality, Ghana (N = 31).

Topic	Educators’ Perspectives	Student Perspectives	Similarities	Differences
**Curricular Approach**	Abstinence Condom use as an option	Abstinence Condom use as an option	Abstinence as a general approach, condom use as an option	
**Content of HIV/AIDS Education**	Causes of HIV/AIDS Modes of transmission and prevention Abstinence and chastity ABC ^1^ approach Stigmatisation	Modes of transmission and prevention Abstinence and chastity	Modes of transmission and prevention Abstinence and chastity	Educators go further to include causes of HIV/AIDS
**Modes of delivery and implementation**	Integrated into regular subjects such as religious and moral education, physical education, social studies, and science Part of co-curricular activities including worship, clubs and talks	Integrated into regular subjects such as Religious and Moral Education, Physical Education, Social Studies, and Science Part of co-curricular activities including worship, posters, talks, school assembly, outreach programs by students	Infused into regular classroom subjects Implemented through co-curricular activities	More in-depth co-curricular activities described by students to include posters, school assemblies, and outreach programs by students
**Organizations involved in HIV/AIDS education**	Health workers–nurses	Health workers-nurses NGO	Nurses are commonly put forward as external actors involved in HIV/AIDS programs	Students went further to include NGOs
**Attitudes about abstinence and comprehensive HIV/AIDS education**	Abstinence best approach CSE ^2^ is effective because, despite abstinence, students still engage in risky sexual behaviours Classroom-based HIV/AIDS programs are good Combination of teaching methods more efficient Participatory approaches are highly effective Small group discussions are very good Teaching CSE implies the promotion of promiscuity	Abstinence best approach Condom use preferred Values enhancement, such as chastity, is a good approach Classroom-based HIV/AIDS programs are good High preference for programs by third parties, such as nurses Group-based activities by teachers are highly preferred Worship and large gatherings are the least effective Highly sceptical approach to the use of condoms	General preference for abstinence Condom use is preferred by both teachers and students Classroom-based programs are generally preferred Group discussions facilitated by teachers are generally preferred	For students, abstinence is the best approach because of condom failure Teachers reported that abstinence is ineffective because they know some students are already sexually active
**Recommended improvements**	Sexuality education as a standalone subject More advocacy on HIV programmes Inclusion of parents	Intensification of HIV/AIDS activities Innovative ways of teaching HIV/AIDS education New teaching and learning materials Individual counselling of students Third-party involvement in programs Proper demonstrations on condom use	General agreement on the need to intensify HIV/AIDS education	Intensification of HIV/AIDS activities recognised through HIV education as a standalone subject by teachers, while students recommended innovation in teaching and condom use demonstration, new teaching and learning materials, and third-party involvement

^1^ ABC—Abstinence, Be faithful, Condom use. ^2^ CSE—Comprehensive Sex Education.

## Data Availability

Not applicable.

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
