# Peer review of "Conflicting HIV/AIDS Sex Education Policies and Mixed Messaging among Educators and Students in the Lower Manya Krobo Municipality, Ghana"

_ijerph, 2022, doi:10.3390/ijerph192315487_

Round 1

Reviewer 1 Report

Thank you for the invitation to review this paper. I have some suggestions that  can be find bellow:

The abstract should be better constructed. It is very confuse.  Please, show the main purpose of the paper, I mean, the originality of the study. Also, the methods should be included in the abstract.In the results you must mention the sample of your study. Please, be more certain in the conclusion. 

In the introduction, I could not see the originality of the study. Where is the gap in the literature that the authors propose to fill? Was this manuscript the first one to address this subject? In the introduction it seems both methods of sex educations studied already. 

Lines 91-96 should be on the topic “study site”

In line 132 when you mention “district” you must insert the name of the district.

Also in the methods, I was concerning if the interview formulary was evaluated by other researchers before application. 

Discussion and results are well done.

Also, be more direct in the conclusion. It is also too long.

In the limitation of the study the authors mentioned the sample size, however  in qualitative studies this is not a problem when the answers starting to converge to a same point. Did it happen in your study?

Reviewer 2 Report

Thank you for the opportunity to review “Student and educator perspectives of HIV/AIDS sex education in Junior High Schools: A qualitative study in Agormanya HIV sentinel site, Lower Manya Krobo Municipality, Ghana”. The study selected 31 students and HIV educators to canvas opinions about HIV prevention education curricula delivered in Ghana.

While I appreciate the intent of this paper is to point out the confusion experienced by students about what appear to be competing curricula (although the authors use “dual modes”, line 84, in fact they are competing), student, teacher and nursing confusion is only a symptom of the failure of government to set clear policy in this area. It would appear that the Ghanian government and its various ministries have failed to have the essential—and possibly life-saving—discussions at the highest levels which would ensure consistent messaging about HIV prevention throughout the country. Instead, government seems to have relegated these discussions to the front lines of HIV education. That is what this paper is really about, using the Krobo Municipality as a kind of case study. The single most important lines in this paper are lines 639-641 (“The study identifies ambiguity in sex education policies as a possible threat…”), but this central message lies buried at the end of the paper. I encourage a revision of the paper that centralises this critical message, places it as the focal thesis at the beginning of the paper, and then uses the existing data to support that thesis. In order to do this, of course, the authors will need to tell us something about the political context of HIV prevention discussions in Ghana, but at the authors themselves remark the literature is replete with studies in Ghana and other parts of SSA (line 46), and this study as it is currently drafted does not add very much that is new to that literature. An examination of how failures of government contribute to the rising infection rates in young people would be a valuable addition. I appreciate that this may be a risky way to frame this article for in-country authors, but I’m sure a way could be devised to make the point clearly and safely. I note that the ABC curriculum is a US-funded curriculum, and its continued use suggests something about larger international relationships and the reliance of Ghana on international funders. This is a point that could be made.

The authors do not locate themselves anywhere in the paper—who are they, and what is their relationship to student participants? What is their relationship with the University of Utah, the funder and ethic reviewer of the study? Who did the actual interviewing of students, and in what language? (Dangbe? Klogbi? Something else?) How were translations managed and checked? Was there member-checking?

We also need more background about the cultural context of this paper, since culture is such an integral part of the author’s argument. We don’t even discover that Krobo is a culture, not merely a municipality, until line 570! I was frankly surprised to see the word ‘natives’ appear no less than five times in reference to the residents of this area of Ghana—to those of us used to notions of indigeneity this word clangs loudly. There is some useful description in section 2.1, but there are problems here: lines 170-171 don’t make sense, the impact of colonisation is omitted, and there are local terms that are unexplained to the international reader (e.g., Queen Mothers, line 191); other cultural components (like clubs) are left to the reader to imagine (see ‘the No Yawa Club’ line 325). There is reference to ‘Christian’ religion throughout, but Christianity is not a single religion: is this Roman Catholic, Pentecostal/Evangelical, Mormon, something else, or some combination of all of these?  Each of these has different understandings and theologies of birth control and condom use. And what is the role of a class on Religion and Moral Education in a state-funded public school?  All of this should be abundantly clear to a reader that has no prior knowledge of East Ghanian culture.

A third major concern for me is some of the ways the paper is written. Quotations appear inconsistently, (single vs. double quotations, sometime italic, sometimes not; this also occurs in titles—see line 69), and there are many, many acronyms—the authors need to keep in mind that they are writing for an international audience, and if they have to keep flipping back to find out what an acronym means they will quickly lose interest. One of the points of writing an article is to get it cited and quoted, and no one is going to quote an article with many acronyms that require clarification. Use plain language (e.g., ‘an abstinence curriculum’, ‘a condom-inclusive curriculum’, etc.) and avoid other local jargon.

The Methodology is unclear. The authors state that their purpose is to ‘describe’ (line 116) but later, expand this to include ‘explain’ (line 121). Describing is observational, explanation requires interpretation. What interpretive methods did the authors use?  Was there member-checking? How did the authors manage social desirability bias in the responses of young people?

Although there is some explanation of the recruitment methods, ‘key informants’ were actually participants. The authors write that peer educators ‘were considered’ for inclusion—but were they in fact included? In line 142ff, ‘Informed consent was administered” is not meaningful: informed consent is obtained from participants, not administered to them. It is not at all clear that student interests were protected by teachers offering approval unless approval for participation in research is something that is specifically delegated to teachers by parents—and the paper does not make that clear. The language and procedures in this section must be very explicit and clear, and they are not.

Finally, there are limitations other than size which the authors do not clearly identify. An adequately done qualitative study does not rely on size—probability sampling is related to quantitative research and generalisability. That is not the authors’ purpose in this study, so this is not a limitation. However, nowhere do the authors address how they manage social desirability bias in young people. Did any of the young boys say that they were sexually active with other men (line 209)? How would that have been heard by the interviewers? How were translations managed and in what language did the analysis occur—how was meaning preserved? These are much more meaningful limitations.

Minor concerns

Title: The title is very long. Can it not be shortened? Consider something like ‘Conflicting HIV policies lead to confusion among HIV educators in Lower Manya Krobo Municipality Ghana’, or something even shorter and better than that. You want to be found and cited.

Abstract, line 12, ‘In contrast’—in contrast to what? The abstract should be revised when the paper is revised to state clearly what the authors’ purpose is in writing the paper.

Line 34, ‘700,000 1new HIV…’ [sic] is unclear

Lines 91 and following are confusing. Consider revising to: ‘In 2018, HIV prevalence in the general population of LMKM was 5.56%, more than twice the national average of 1.69%. In 2018 prevalence in young people ages 15-24 was 0.8%, the highest in the nation. Incidence in young people increased from 5.30% in 2013 to 5.56% in 2016.’

Line 109, the six principles should be in a parenthetical or this line should be revised altogether. They don’t belong after a colon, and this material is never referred to again. Is it really necessary?

Line 226, what is ‘living on their cover’?

Line 235 repeats material presented earlier.

Line 356, ‘even put it over-caution of sharp objects as a mode of contraction of the virus’. What does this mean?

 Table 2 has so much information in it and is so long I’m not sure that it is useful. A table is meant to summarise complex information, and all this one does is re-present it in a table form. Consider omitting it, and simply make the case in the text. Ask this question: Is this table useful for anyone else other than the authors?

Reviewer 3 Report

This is an interesting paper examining student and educator perspectives of HIV/AIDS sex education in Ghana using a qualitative research method. The research method adopted is reasonable and the findings are interesting.

1. As a qualitative study, the authors have to clarify the qualitative orientation of the study. In particular, what qualitative elements were incorporated in the study?

2. How did the researchers control the ideological biases and preoccupation in the study?

3. I encourage the authors to discuss the findings in a more self-critical manner. 

The authors will find the following paper useful:

Shek, D. T. L., Tang, V. M. Y., & Han, X. Y. (2005). Evaluation of Evaluation Studies Using Qualitative Research Methods in the Social Work Literature (1990-2003): Evidence That Constitutes a Wake-Up Call. Research on Social Work Practice, 15(3), 180–194. https://doi.org/10.1177/1049731504271603

Round 2

Reviewer 2 Report

Thank you for the opportunity to re-review this paper, now titled ‘Conflicting HIV/AIDS sex education policies and mixed messaging among educators and students in the Lower Manya Krobo Municipality, Ghana. I appreciate and acknowledge the authors’ openness to feedback on both substance and style, and I think the paper is much stronger and clearer for the changes. I like the new title very much. The paper more clearly elucidates the challenges in the HIV prevention environment that both teachers/adults and students are coping with: perhaps this paper will go some way to encouraging politicians to address this problem.

When I refer to ‘location of the authors’ I do not mean their specific role in the research project: that is set out in the note at the end of the paper. What I’m asking about is why the authors undertook this study to begin with: what was their point of view? Was this commissioned research, or something that they felt passionately about? This passion informs the motivation and possibly point of view of the research. Locating the researchers helps readers to understand more about the context of the researcher, and helps readers assess any researcher perspectives in the interpretation of the findings. I don’t need yet another revision on this minor point, but it is always important in qualitative studies—which rely on multiple hermeneutic steps, particularly when there are several languages involved—to situate the research and the researchers. I think it is very, very interesting that students chose to engage in English, which somehow makes the entire topic of ‘sex’ more distant from their own cultural experiences. It is easier to talk about difficult things in a language that is not my own because words lose some of their power when they are in a second language. Such things are so automatic that we often don’t stop to think about why they are happening. I wonder if the findings would have been the same if the interviewer had used only Krobo? That would have made it very personal. This again is an observation for the authors to consider, not a requirement to amend.  

I have a few minor editing suggestions for the authors. These are not requirements (although some are typographical and referencing issues that should be addressed), but are points for the authors to consider in their final review:

Line 67: Since the point here is that curriculum design is political as well as technical, perhaps this phrase could be revised to ‘CSE curriculum design is a political as well as a technical process’. The reference could stand as it is since this is not a direct quotation.

Line 79 (and elsewhere): I believe the abbreviation is ‘USAID’ without the parentheses.

Line 154: Key informants are different from participants, at least in my research experience; however, the point is a minor one, and as long as it is clear to the reader, I don’t need to pursue this point.

Lines 255-256, there is either an extra or missing parenthetical. I would simply say ‘Christian (Catholic, Anglican, Lutheran, and Pentecostal)’. I don’t think ‘charismatic’ is the name of a denomination, but an adjective to describe the worship style of any of these sects. However, I don’t know the situation on the ground in Ghana, and if some churches name themselves as ‘Charismatic’ (as a noun), then by all means include it as such, but as a separate denomination.

Line 384—missing space between ‘abstinence’ and ‘sex’.

I yield to the authors’ clearly stated desire to include Table 2, but it is huge, and I wonder if reducing the line spacing would at least keep it to two pages rather than overrunning to three. Column headings should be repeated on the second page.

Line 225-6 (the second series on p.19), I wonder if the authors would consider rewording this to ‘due to the cultural tendency of students to present themselves…’. It is the tendency that is cultural.

I acknowledge the addition and revision of referencing throughout. However, a number of these references in the reference list are incomplete, missing volume and/or issues numbers (e.g., 9, 10). References 26 and 39 (for example) are inconsistently formatted. In other words, it would be good to have an editor carefully review the entire reference list for completeness, format, and consistency.
